# Of Flies and Men—The Discovery of TLRs

**DOI:** 10.3390/cells11193127

**Published:** 2022-10-05

**Authors:** Hauke Johannes Weiss, Luke Anthony John O’Neill

**Affiliations:** School of Biochemistry and Immunology, Trinity Biomedical Sciences Institute, Trinity College Dublin, D02 R590 Dublin, Ireland

**Keywords:** Toll-like receptors (TLRs), innate immunity, immunology, Drosophila, LPS, innate immune signaling, Nobel Prize, TIR domain

## Abstract

In 2011, the Nobel Prize in Physiology or Medicine was awarded to three immunologists: Bruce A. Beutler, Jules A. Hoffmann, and Ralph M. Steinman. While Steinman was honored for his work on dendritic cells and adaptive immunity, Beutler and Hoffman received the prize for their contributions to discoveries in innate immunity. In 1996, Hoffmann found the *toll* gene to be crucial for mounting antimicrobial responses in fruit flies, first implicating this developmental gene in immune signaling. Two years later, Beutler built on this observation by describing a Toll-like gene, *tlr4*, as the receptor for the bacterial product LPS, representing a crucial step in innate immune activation and protection from bacterial infections in mammals. These publications spearheaded research in innate immune sensing and sparked a huge interest regarding innate defense mechanisms in the following years and decades. Today, Beutler and Hoffmann’s research has not only resulted in the discovery of the role of multiple TLRs in innate immunity but also in a much broader understanding of the molecular components of the innate immune system. In this review, we aim to collect the discoveries leading up to the publications of Beutler and Hoffmann, taking a close look at how early advances in both developmental biology and immunology converged into the research awarded with the Nobel Prize. We will also discuss how these discoveries influenced future research and highlight the importance they hold today.

## 1. Introduction

The 2011 Nobel Prize to Hoffmann, Beutler, and Steinmann was a testament to the advances in research on innate immunity made in the 1990s and early 2000s. While concepts of adaptive immunity were already well described at the time, the function of innate immune cells was mostly thought to be reduced to the activation of the adaptive immune system, the nonspecific elimination of microbes and materials by phagocytosis, and inflammation. While Steinmann bridged the gap between the two with his work on dendritic cells [1,2,3], it was the work of Hoffmann and Beutler that elevated the field of innate immune recognition. Hoffmann published on the antimicrobial function of the developmental gene *toll* in *Drosophila melanogaster* in 1996 [4], and Beutler built on these data by identifying the Toll-like receptor TLR4 as the receptor for bacterial lipopolysaccharide (LPS) in mammals in 1998 [5], sparking a revolution in the field. These publications gave rise to the identification of the TLR family of receptors and allowed for the discovery of many other pattern recognition receptors (PRRs) as well. While ground-breaking, the findings of Hoffmann and Beutler were built on decades worth of discoveries in the fields of immunology and developmental biology, the most important of which we will discuss in this review.

## 2. Toll and Pathways in *Drosophila*

The origin of Toll in immunity can be traced to development biology research. The fruit fly *Drosophila melanogaster* has been a key model organism in studying developmental biology and genetics for over a century, and, besides Hoffmann, four more Nobel Prizes have been awarded to scientists for their work with this animal. The first experiments involving *Drosophila* were conducted as early as 1901 by William Castle at Harvard University [6]; however, it was Thomas Hunt Morgan who achieved the first breakthrough results with *Drosophila*, confirming Mendel’s theory of inheritance and establishing the concepts of genes and chromosomes [7]. This discovery earned him the 1933 Nobel Prize in Physiology or Medicine. Morgan’s student, Hermann Muller, later found that radiation mutates genes and that the offspring of irradiated *Drosophila* have mutated genes too [8], earning him a Nobel Prize in physiology or medicine as well. Easy to breed and inexpensive to house, *Drosophila* quickly became the gold standard animal model in genetic research.

Modern *Drosophila* research emerged in the 1970s, when new techniques allowed for the genetic mapping of embryonic development. It emerged that larvae contain pre-determined groups of cells—so-called imaginal discs—that would give rise to the various features and body parts of the adult *Drosophila* yet hold no function in the larvae themselves [9,10]. These imaginal discs could be transplanted [11,12], allowing for the investigation of their individual functions. Clonal analysis of these discs and the resulting labelling techniques [10,13] helped in understanding the fate of these individual cell clusters, resulting in the discovery of anterior (A) and posterior (P) compartments within various structures of the adult fly, being based on different imaginal discs [14,15,16]. Several morphogens that organize the development of distinct compartments were discovered during this time [17]; among them is *dorsal*, a mutation which caused a fully dorsalized phenotype [18]. Other mutants displaying similar phenotypes, including *pelle*, *tube*, *toll*, *spätzle*, and *cactus* [19,20], which soon emerged one after the other, revealed the genetic blueprint of the animal. A key factor in the discovery of these genes was the utilization of new molecular techniques in DNA cloning [21], and these findings were later awarded with another Nobel Prize to Edward B. Lewis, Christiane Nüsslein-Volhard, and Eric F. Wieschaus in 1995 for their discoveries regarding “the genetic control of early embryonic development” [22].

*Toll* was identified as one of the most important genes in embryonic development and the establishment of embryonic polarity. While most mutations of genes from the *dorsal* group retain some residual polarization, *toll* mutants lack any organization, and dorsoventral polarization by the reintroduction of wildtype (WT) cytoplasm is fully dependent on where the cytoplasm is injected [23]. It was suggested that the *toll* protein product is constantly expressed but would exist in an activated or an inactivated state and that members of the *dorsal* group, including *toll*, might be part of a signaling pathway [24], hypotheses that should later be confirmed on a biochemical level. The *toll* gene product Toll was soon identified as a membrane-bound receptor [25], capable of signal transduction through an extracellular stimulus [26]. This ligand was then identified as Spätzle, another member of the *dorsal* group [27], and Dorsal itself was shown to be a transcription factor recruited to the nucleus upon Spätzle binding to Toll. As stated above, Toll was then shown to be homologous to the mammalian type 1 IL-1 receptor (IL1R1) [28], while Dorsal was closely related to the IL-1-induced transcription factor NF-κB [29]. More similarities between the Toll pathway and the inflammation-inducing anti-microbial IL-1 receptor-induced signaling cascades soon became apparent, laying the groundwork for the discoveries of Hoffmann. Most notably, cactus was shown to be a homologue of the inhibitor of NF-κB (I-κB) in its function to retain Dorsal or NF-κB, respectively, in the cytoplasm [30,31], and both pathways involve structurally related protein kinases, such as the *pelle* gene product (pll) in the Toll pathway [32] and the interleukin-receptor associated kinase (IRAK) in the IL-1 pathway, respectively [33]. In addition, both *dorsal* itself as well as the closely related *Dorsal-related immunity factor* (*dif*) had been shown to exert antimicrobial functions in fat body cells from *Drosophila* [34,35].

Interestingly, a protein structurally related to both Toll and IL1R1 was discovered in a different kingdom of life. The N protein from the tobacco plant shared a cytoplasmic domain with Toll and IL1R1, a domain termed the Toll-IL-1 receptor resistance (TIR) domain by Barbara Baker, who uncovered the role of the N protein in the antiviral responses against *Tobacco mosaic virus* [36]. The developmental role of Toll and its homology to both IL1R1 and the N protein were therefore important precursors to the discovery of the role of Toll in *Drosophila* innate immunity.

## 3. *Toll* Has Antimicrobial Functions

In 1996, Jules Hoffmann published on the antimicrobial function of three different genes, *spätzle*, *cactus*, and *toll* (*tl*), in the fruit fly *Drosophila melanogaster* [4]. Jules Hoffmann was born in Luxembourg on 2 August 1942. From an early age, he had a fascination for insects, an interest that would carry over into his research career. He obtained his doctoral degree in biology in 1969 from the University of Strasbourg, becoming a research associate in the same year. He went on to pursue postdoctoral training at the Philipps University of Marburg, Germany in the area of biochemistry before returning to Strasburg in 1974 to set up his own lab and establish a research unit on the immune response in insects in 1978. Here, he laid the groundwork for the research that would result in him being awarded the Nobel Prize decades later. After initial success working on grasshoppers [37], he turned towards *Drosophila melanogaster* later, identifying antimicrobial polypeptides such as diptericin [38] or defensin [39] as being part of the immune system of these insects. His groundbreaking and Nobel Prize-awarded work, however, was for the 1996 paper on Toll in *Drosophila* immunity, where he and Bruno Lemaitre were the first to describe the Toll pathway as being responsible for the production of the antifungal peptide dorsomycin [4], a discovery sparking a huge interest in similar proteins in mammals and ultimately leading to the discovery of the role of TLRs in innate immunity.

*Spätzle*, *cactus*, and *toll* are part of the dorsoventral signaling pathway and, curiously, the activation of the pathway had first been shown to lead to the activation of the dorsoventral morphogen *dorsal*, which is key to the establishment of dorsoventral polarity in the developing *Drosophila* embryo [40]. Hoffmann found that all three of these genes are involved in mounting an antifungal response, showing a decreased induction of the antifungal peptide gene *drosomycin* in animals with mutations in *spätzle*, *cactus*, or *toll*. *Dorsal*, however, was not involved. Drawing parallels to the components of the NF-κB pathway, Hoffmann described how the activation of the Toll receptor by its ligand spätzle leads to a similar signaling cascade as the activation of IL1R1 by IL-1 [41]. As stated above, IL1R1 had been shown to be a homologue to Toll, as well as the N protein in tobacco [36], which led to the concept of the ‘TIR’ domain. The N protein had been shown to be required for resistance to *Tobacco mosaic virus*. That Toll had been shown to be involved in host defense made biological sense, since IL1R1 and the N protein were already known to be similarly involved in immunity in mammals and plants, respectively. The TIR domain was therefore a very ancient signaling domain for innate immunity stretching back billions of years to the common ancestor of plants and animals.

Toll drove an antifungal response but not antibacterial response. Antibacterial responses are facilitated by the *immune deficiency* (*imd*) gene [42], with the antibacterial peptides cecropin, attacin, and defensin being partly dependent on Toll and diptericin and drosocin being independent of Toll. These results were achieved by the mutation of components of either the *imd* or *tl* pathways and the subsequent investigation of the downstream activation of components of these antimicrobial pathways or the challenge of the mutated flies with bacterial or fungal infections. This paper was therefore the first to describe the Toll pathway as a component of the innate immune system, a finding that will later be expanded to mammals and lead to the discovery of a multitude of receptors and pathways in innate immunity common to plants and animals.

## 4. Innate Immune Signaling and LPS

The idea of innate immune sensing by so-called pattern recognition receptors was proposed by Charles Janeway in 1989 [43]. TLR4 emerged as a prototypical PRR. It is worth going back a bit further in time to fully understand the motivations that drove Beutler in his pursuit of the LPS receptor. LPS was first described as an “endotoxin” derived from *Vibrio cholerae*, the bacterium responsible for cholera, by Richard Pfeiffer in 1892, inducing fever and death even when the bacterium was killed [44]. Soon after, Eugenio Centanni isolated the substance and proposed it to be non-proteinous due to its remarkable heat stability [45]. Around the same time, physician William B. Coley started treating his cancer patients at the Memorial Hospital in New York with heat-inactivated bacteria, observing tumor remission in some cases [46], which, today, we know was due to the strong activation of the immune system by the bacterial LPS [47]. The endotoxin was identified as lipopolysaccharide in 1943 [48] but was not fully structurally identified until the 1980s [49], and while the detailed composition and the strength of the response may vary depending on the bacteria in question, they all induced the same inflammatory symptoms [50]—that is, until the 1960s. Suddenly a mouse strain was mentioned that was resistant to LPS-induced inflammation [51], indicating how this susceptibility to endotoxin may be down to very distinct cellular processes. It was later discovered that the resistance stemmed from a mutation in a distinct locus, appropriately named *lps* [52,53]. These mice, as well as another resistant strain identified in 1977 [54], termed C57BL/10ScCr, were important pillars for Beutler’s work, as the genetic analysis of the *lps* locus would provide the key for the discovery of the LPS receptor.

With emerging knowledge about the individual cells acting within inflammation and the immune system, it became clear that macrophages were the main cell type responsible for LPS-induced inflammation [55], raising the question of whether such an intense reaction to LPS was a good thing or a bad thing. As a strong inducer of sepsis, it might be viewed as a purely toxic substance; however, Coley’s work showed how LPS responses could keep malignancies at bay. In addition, by the end of the 1960s, it was clear that LPS could be beneficial in pathogen clearance, acting as an adjuvant [56,57,58]. In the 1980s, it was also shown that mice of the above-mentioned strains that lack an LPS response are more susceptible to infection with gram-negative bacteria [59,60], the very organisms that carry LPS on their surface. It was therefore clear that a sensing mechanism for LPS must exist to detect and respond to LPS-bearing bacteria. One of these responses included the production of cytokines such as tumor necrosis factor (TNF) [61], the induction of which was found to be at least partly responsible for LPS-induced toxicity [62], and TNF seemed to be beneficial in resolving the same type of microbial infections as LPS [63,64].

The work on cytokines such as IL-1 and TNF, which overlap in many of their pro-inflammatory effects, in turn provided the first ideas as to how the signal transduction from LPS to the nucleus might work. In the years before the discovery of TLR4, it had already been established that IL-1 binds IL1R1, resulting in the activation of the transcription factor NF-κB, as does LPS [65]. Two LPS-binding proteins could be identified prior to the discovery of TLR4. The LPS-binding protein (LBP) was first described in 1986 [66] and was found to enhance LPS-induced signaling events [67,68]. This protein, however, was not membrane-bound and could thus not act as a signal transducer. The second receptor, CD14, is a membrane-bound protein [69,70], but it lacks cytoplasmic domains [71]. Then, in 1996, Hoffmann published his work on the antimicrobial role of Toll in *Drosophila* [4], and, suddenly, the relevance of this receptor family in immunity became apparent and gave Beutler a reason to pursue TLR4 once it came up as a candidate in the unravelling of the *lps* locus. Importantly, Charles Janeway and Ruslan Medzhitov had shown in 1997 that TLR4, described as human Toll in their paper, could activate immune cells [72]. While other TLR members had been described before TLR4, their function and ligands remained unknown until after the identification of TLR4 as the LPS receptor [73,74], further highlighting the significance of Beutler’s discovery. The most important milestones leading up to Beutler and Hoffmann’s discoveries are listed in a timeline in Figure 1.

## 5. TLR4 Is the Receptor for Bacterial LPS

The search for the LPS receptor started with a thorough investigation of the so-called *lps* locus after it was reported decades previously that C3H/HeJ mice, a strain with mutated *lps*, are unresponsive to LPS [75,76]. LPS resistance was also shown in mice of the C57BL/10ScCr strain [54,77]. While an LPS-binding protein termed cluster of differentiation 14 (CD14) had already been identified on the surface of macrophages, this protein is not able to transduce a signal due to a lack of a cytoplasmic domain, as described above [70]. CD14 augments LPS-induced signaling [71]; however, it was likely to act as a co-receptor for the then-unknown TLR4. Utilizing genetic and physical mapping data, Beutler characterized the *lps* locus and identified *tlr4* in the area mapped. Bruce A. Beutler was born on 29 December 1957 in Chicago, Illinois. Growing up in California, he graduated from the University of California, San Diego in 1976 at only 18 years old with a degree in biology. Beutler then pursued an M.D. degree at the University of Chicago, which he obtained in 1981. After postdoctoral training at Rockefeller University in New York, he became Assistant Professor in 1985 before moving to Dallas a year later. It was in New York where he made one of his first important discoveries: isolating tumor necrosis factor-alpha (TNF-α) in mice [61]. At the University of Texas (UT) Southwestern Medical Center in Dallas, he obtained an associate professorship in 1990 and, finally, a full professorship in 1996. Here, he continued working on TNF, generating Immunoglobulin G (IgG) heavy chain-based TNF antagonists [78], eventually leading him to become interested in one particular TNF-inducing agent, LPS. Setting out to explain the induction of TNF and other cytokines by LPS, together with Alexander Poltorak, he was able to identify Toll-like receptor 4 (TLR4) as the receptor for LPS in 1998 [5], earning him his Nobel Prize.

*Tlr4* was a promising target, as it was by then known to be part of the IL1R/TLR family. Comparing the macrophage mRNA and genomic DNA from an LPS-responsive vs. C3H/HeJ mouse strain, Beutler and collogues identified a mutation in the *tlr4* gene, resulting in a proline to histidine substitution in the *tlr4* gene from the LPS-unresponsive mice. In the C57BL/10ScCr strain, *tlr4* mRNA was entirely absent, further supporting the importance of this gene in LPS-induced signaling. Finally, TLR4 was shown to be downregulated in response to LPS treatment, possibly to facilitate tolerance to LPS, a well-known phenomenon. In the context of the discoveries made by Hoffmann two years prior [37], Beutler therefore concluded that TLR4, like the *Drosophila* homologue Toll, launched a response to microbial infection. In the case of TLR4, however, this pathway has evolved to detect gram-negative bacteria, with developmental aspects that had been described for Toll in insects being lost. The homologies between Toll and TLR4 are highlighted in Figure 2. In this work, Beutler expands the discovery made by Hoffmann in insects to mammals, revealing one of the most important antimicrobial receptors in mammals in the process.

The Nobel Prize was awarded to Hoffmann and Beutler in recognition of their contribution to the discovery of Toll and TLRs in innate immunity. While years of research are represented in this award, these two papers are particularly important.

## 6. Scientific Impact

After the function of TLR4 was determined, the field of innate immune recognition exploded. Suddenly, the innate immune system was no longer seen as a crude, unspecific system with the simple task of activating adaptive immunity. It proved to be a sophisticated detection system capable of responding to and selecting between distinct microbial threats.

Soon after TLR4, the functions of many other TLRs were defined, each with another specific microbial ligand. Shizuo Akira found that TLR2 and TLR6 recognize gram-positive bacteria by binding to lipid structures [79,80], and TLR9 was shown to bind to bacterial DNA [81]. TLR3 was found to bind to double-stranded RNA (dsRNA), as found after viral infection [82], TLR5 was found to bind bacterial flagellin [83], and TLR7 and 8 were found to recognize viral single-stranded RNA (ssRNA) [84]. TLR10 binds triacylated lipopeptides [85] and is expressed in humans but not in mice. Conversely, mice express TLR11, which binds bacterial profilin and flagellin [86,87], TLR12, which binds profilin [86], and TLR13, which binds bacterial ribosomal RNA [88], which are all missing in humans. In addition, TLR2 was found to heterodimerize with TLR1 or TLR6 [89], while TLR11 dimerizes with TLR12 [90]. In Figure 3, all currently known TLRs are displayed, with their ligands and primary adaptors.

Besides TLRs, other PRRs were also discovered in the following years—most notably, NOD-like receptors (NLRs) [91], the first of which was discovered in 1999 [92], RIG-like receptors (RLRs) [93], first described in 2004 [94], and C-type lectin receptors (CLRs) [95], which, although being considered part of the immune system for a long time, were shown to bind microbial products in 2000 [96].

There were also remarkable advances in determining the signaling pathways initiated by TLRs. Adaptor proteins involved in signaling were shown to have TIR domains. The first to be described was Myeloid differentiation primary-response protein 88 (MYD88). Discovered before the TLRs, it was found to bind to IL1R1, facilitating downstream signaling events [97], and, subsequently, it was found to mediate TLR signaling in TLRs as well [98]. Signaling was therefore initiated by TIR–TIR interactions from the receptor to adaptors. An exception was TLR3, which signals through TIR domain-containing adaptor protein inducing IFNβ (TRIF) [99,100]. TLR4 is also able to signal independently of MYD88 through TRIF when endocytosed [101,102]. TRIF-related adaptor molecule (TRAM) recruits TRIF to the TLR signaling complex [103,104], while MYD88-adaptor-like protein (MAL) recruits MYD88 to TLR4 and all other TLRs that use MYD88 [105]. From here on, TLR signaling gets more complex (described in more depth elsewhere) [106,107]. Even today, new aspects of TLR signaling are still being uncovered, especially revolving around the functionality of TIR domains. The TIR domain containing protein sterile alpha and the TIR motif containing 1 (SARM1), for example, were found to have enzymatic activity, depleting nicotinamide adenine dinucleotide (NAD^+^) [108]. Indeed, several TIR domains have been found to have NADase activities, across various species as well [109]. While TIR domains from TLR signaling components other than SARM1 in animals do not seem to have NADase activity, the discovery of the enzymatic activity of this domain gives this aspect of TLR-signaling molecules a whole new angle and contributes to the burgeoning field of immunometabolism.

## 7. Concluding Remarks

Overall, it cannot be denied that the discoveries that resulted in the Nobel Prize being awarded to Hoffmann and Beutler are among the most important immunological milestones of the outgoing 20th century and are a testament to the tenacity of both scientists, along with the members of their research groups. They reshaped the way innate immunity was perceived and our understanding of how the first line of defense in our body actually works. While the Nobel Prize was well deserved, other pioneers in the field deserve recognition too. Without the groundbreaking discoveries of Nüsslein-Volhard in the field of Toll signaling or the emphasis placed on the innate immune system by Charles Janeway, these discoveries would not have been possible. Additionally, other brilliant scientists, such as Ruslan Medzhitov and Shizuo Akira, should be mentioned alongside Hoffmann and Beutler for their many contributions. The two key publications described here truly elevated the field, the consequences of which are still being extensively explored by many scientists.

## Figures and Tables

**Figure 1 cells-11-03127-f001:**
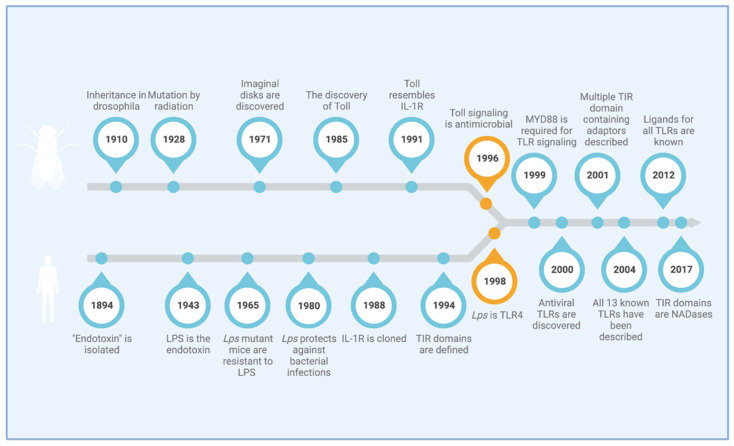
Milestones along the way to TLR discovery. The highlighted publications contributed significantly to today’s knowledge on TLRs and innate immunity.

**Figure 2 cells-11-03127-f002:**
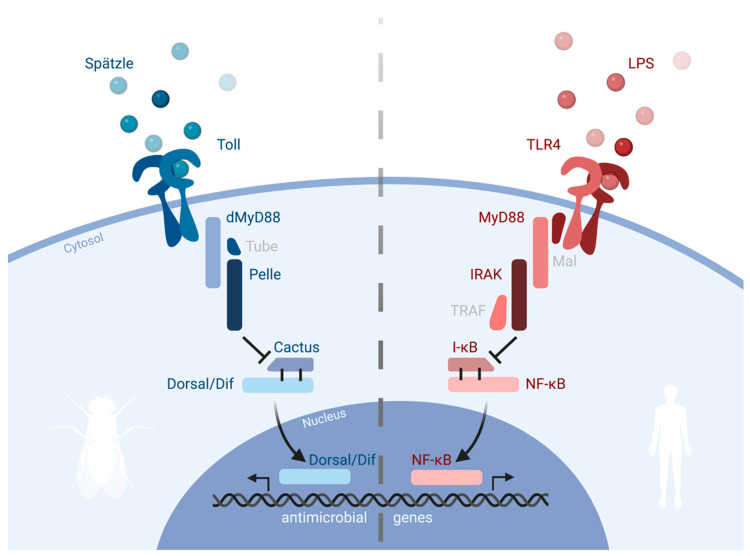
Simplified illustration of the Toll and TLR4 signaling pathways. Homologue components of both pathways are depicted in blue and red, respectively.

**Figure 3 cells-11-03127-f003:**
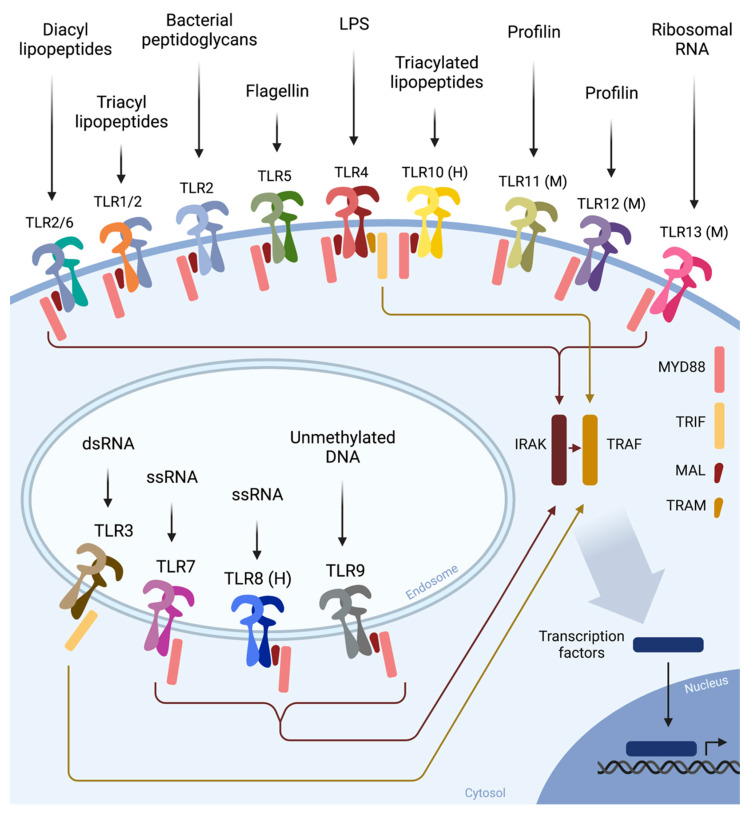
Illustration of all known TLRs and their ligands and adaptors, including those exclusive to mice (M) and humans (H).

## Data Availability

Not applicable.

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
