# Peer review of "Of Flies and Men—The Discovery of TLRs"

_cells, 2022, doi:10.3390/cells11193127_

Round 1

Reviewer 1 Report

Twenty five years after key discoveries and 11 years after the Nobel Prize award to some of their authors, Weiss and O’Neill commemorate the discovery of TLRs by highlighting milestone works by the Beutler and Hoffmann labs. The narrative is effective, the authors have themselves the authority in the field to support such acknowledgement, and their appreciations are objective. The article is not yet another TLR review, but a didactic account on how TLR signaling was uncovered, bringing to the forefront the role of model organisms, such as Drosophila, paying tribute to the scientists who contributed to boost the field, and providing historic must-read text for newcomers to the area of Innate Immunity and signaling. The article reviews previous essential work both in flies and mice or immune cells related to LPS detection, that was the seed of these ground-breaking discoveries in the ’90s, then focuses on the relevance of the findings, to conclude with a brief account of important results derived, such as the description of the TLR family and the signaling mechanisms. The article should be published. Some suggestions follow:

Major point

The only figure of the manuscript, showing a milestones timeline, is very informative for the purpose of the paper. However, other figures should help the naïve reader to interpret the pathways described. As the article has a special interest for reader who approach the field for the first time, extra figures illustrating key aspects of the text would be desirable. This said, it can be a challenge for the authors to produce figures that are not repetitive to those in other reviews or, even, in text books. Some ideas are: a figure illustrating the parallelism of the Drosophila Toll and mammalian TLR4 as described in section 2, and a figure showing the repertoire of TLRs (plus respective adaptors) in murine and human TLRs described in Section 6. This reviewer ignores how to make those figures different from published material, but they would help interpreting the points.

Minor points

The address is duplicated. Both authors are form the same institution

Drosophila in title 2 should be in italics

Line 142: “Insect beeing” should be “insect being”

Line 279: comolex should be “complex”

The reference list should be edited for format homogeneity. Note, for instance, that Science is cited as “Science (80-“

Author Response

REFEREE 1

Twenty five years after key discoveries and 11 years after the Nobel Prize award to some of their authors, Weiss and O’Neill commemorate the discovery of TLRs by highlighting milestone works by the Beutler and Hoffmann labs. The narrative is effective, the authors have themselves the authority in the field to support such acknowledgement, and their appreciations are objective. The article is not yet another TLR review, but a didactic account on how TLR signaling was uncovered, bringing to the forefront the role of model organisms, such as Drosophila, paying tribute to the scientists who contributed to boost the field, and providing historic must-read text for newcomers to the area of Innate Immunity and signaling. The article reviews previous essential work both in flies and mice or immune cells related to LPS detection, that was the seed of these ground-breaking discoveries in the ’90s, then focuses on the relevance of the findings, to conclude with a brief account of important results derived, such as the description of the TLR family and the signaling mechanisms. The article should be published. Some suggestions follow:

Thank you very much for your kind words. We are delighted you found our review suited for publication.

Major Point

The only figure of the manuscript, showing a milestones timeline, is very informative for the purpose of the paper. However, other figures should help the naïve reader to interpret the pathways described. As the article has a special interest for reader who approach the field for the first time, extra figures illustrating key aspects of the text would be desirable. This said, it can be a challenge for the authors to produce figures that are not repetitive to those in other reviews or, even, in text books. Some ideas are: a figure illustrating the parallelism of the Drosophila Toll and mammalian TLR4 as described in section 2, and a figure showing the repertoire of TLRs (plus respective adaptors) in murine and human TLRs described in Section 6. This reviewer ignores how to make those figures different from published material, but they would help interpreting the points.

Thank you for the suggestions. We agree that the article would benefit from additional figures. Building on your suggestions, we have added two figures, one depicting the homologies between the Toll and TLR4 pathways, and one summarizing all known TLRs, their ligands, and their primary adaptors.

Minor points

The address is duplicated. Both authors are form the same institution

Drosophila in title 2 should be in italics

Line 142: “Insect beeing” should be “insect being”

Line 279: comolex should be “complex”

The reference list should be edited for format homogeneity. Note, for instance, that Science is cited as “Science (80-“

These points have been addressed in the respective lines. The bibliography has been edited for consistency. All changes have been tracked using the MS Word tracking function.

Reviewer 2 Report

The work is an important contribution to this specific field of research.

It is a work that is thoroughly described and well structured, and there are suitable and convenient references to related works. The previous study is also interesting.

Finally, in my opinion, the work is scientifically sound, and includes a brief review of recent history between the lines.

Author Response

REFEREE 2

The work is an important contribution to this specific field of research.

It is a work that is thoroughly described and well structured, and there are suitable and convenient references to related works. The previous study is also interesting.

Finally, in my opinion, the work is scientifically sound, and includes a brief review of recent history between the lines.

Thank you very much for your kind words. We are delighted you found our review suited for publication.

After a suggestion from another reviewer, we have added two figures, one depicting the homologies between the Toll and TLR4 pathways, and one summarizing all known TLRs, their ligands, and their primary adaptors. Additionally, some minor changes have been made to correct spelling errors. All changes have been tracked using the MS Word tracking function.

Reviewer 3 Report

Weiss and O’Neill presented an interesting review on the discovery of TLRs. The MS title as well as the general introduction and abstract on TLRs which gives the impression the MS will approach all TLRs, but instead, the review is focused on the TLR4 and its importance as LPS sensor. 

If the authors objectives were to approach this receptor in particular, I suggest them to make this clearer and also provide in depth information on all which is known about the receptor. A suggestion would be to talk about homology and similarities between human and other species. Also, TLR4 was discovery to play an important role in SARS-CoV-2 recognition and this should also be discussed. Figures should be added to the review to illustrate and better present the story of TLRs. 

If their objective was to discuss the role of all TLRs they should improve the review to approach their importance as sensors of different pathogens. Nonetheless, TLRs are important for much broader physiological responses which go beyond pathogen recognition. I suggest the authors to make clear the focus of their review.  

Author Response

REFEREE 3

Weiss and O’Neill presented an interesting review on the discovery of TLRs. The MS title as well as the general introduction and abstract on TLRs which gives the impression the MS will approach all TLRs, but instead, the review is focused on the TLR4 and its importance as LPS sensor. 

If the authors objectives were to approach this receptor in particular, I suggest them to make this clearer and also provide in depth information on all which is known about the receptor. A suggestion would be to talk about homology and similarities between human and other species. Also, TLR4 was discovery to play an important role in SARS-CoV-2 recognition and this should also be discussed. Figures should be added to the review to illustrate and better present the story of TLRs. 

If their objective was to discuss the role of all TLRs they should improve the review to approach their importance as sensors of different pathogens. Nonetheless, TLRs are important for much broader physiological responses which go beyond pathogen recognition. I suggest the authors to make clear the focus of their review.

Thank you very much for your constructive comments. This review is indeed focussed on the research leading up to the discovery of TLR4 and the resulting Nobel Prize awarded to Hoffmann and Beutler in 2011, rather than a review of all TLRs. It is part of a special issue titled: “In Honor of Nobel Prize in Physiology or Medicine: Cell-Mediated Immunity”, we hope this context will make the approach of the review clearer.

To further underline the focus on the discovery of this specific TLR, we have added a figure specifically dedicated to the homologies between TLR4 in humans and Toll in Drosophila. We have also proposed a graphical abstract that will make the focus of the review more clear.

Additionally we have added another figure summarizing all known TLRs, their ligands, and their primary adaptors in both humans and mice, as was suggested. All changes have been tracked using the MS Word tracking function.

In light the positive comments of two other reviewers to our manuscript, we would prefer to not make major changes to the structure of this review. We believe that including too many new references would take away from the initial discoveries made by Hoffmann and Beutler and from the goal of this review and the special issue it is part of.

Round 2

Reviewer 3 Report

The authors have made positive changes in their manuscript, including 02 novel figures, one depicting the homology between insect and mammalian pathways for TLR4 and the other showing the different TLRs, their ligands and overall pathways. 

However, English revision is still necessary in order to accept the manuscript. 

Author Response

Thank you for considering our revised manuscript. We are delighted you approve of our changes to the article.

In the updated version, we have made a few minor changes to spelling and grammar for continuity and better readability, as was suggested by you.

All changes have again been tracked using MS Word.